# The Classification Method Study of Crops Remote Sensing with Deep Learning, Machine Learning, and Google Earth Engine

**Jinxi Yao [1,2], Ji Wu [3], Chengzhi Xiao [1], Zhi Zhang [1,* ] and Jianzhong Li [4]**

1. Institute of Geophysics and Geomatics, China University of Geoscience, Wuhan 430074, China; 20656yjx@cug.edu.cn (J.Y.); 1202020648@cug.edu.cn (C.X.)
2. Key Laboratory of the Northern Qinghai−Tibet Plateau Geological Processes and Mineral Resources, Xining 810300, China
3. State Key Laboratory of Information Engineering in Surveying, Mapping and Remote Sensing, Wuhan University, Wuhan 430079, China; wuji1996@whu.edu.cn
4. Research Center of Applied Geology of China Geological Survey, Chengdu 610036, China; wanglang@mail.cgs.gov.cn
* Correspondence: 3slab@cug.edu.cn; Tel.: +86-135-4527-3841

**Abstract:** The extraction and classification of crops is the core issue of agricultural remote sensing. The precise classification of crop types is of great significance to the monitoring and evaluation of crops planting area, growth, and yield. Based on the Google Earth Engine and Google Colab cloud platform, this study takes the typical agricultural oasis area of Xiangride Town, Qinghai Province, as an example. It compares traditional machine learning (random forest, RF), object-oriented classification (object-oriented, OO), and deep neural networks (DNN), which proposes a random forest combined with deep neural network (RF+DNN) classification framework. In this study, the spatial characteristics of band information, vegetation index, and polarization of main crops in the study area were constructed using Sentinel-1 and Sentinel-2 data. The temporal characteristics of crops phenology and growth state were analyzed using the curve curvature method, and the data were screened in time and space. By comparing and analyzing the accuracy of the four classification methods, the advantages of RF+DNN model and its application value in crops classification were illustrated. The results showed that for the crops in the study area during the period of good growth and development, a better crop classification result could be obtained using RF+DNN classification method, whose model accuracy, training, and predict time spent were better than that of using DNN alone. The overall accuracy and Kappa coefficient of classification were 0.98 and 0.97, respectively. It is also higher than the classification accuracy of random forest (OA = 0.87, Kappa = 0.82), object oriented (OA = 0.78, Kappa = 0.70) and deep neural network (OA = 0.93, Kappa = 0.90). The scalable and simple classification method proposed in this paper gives full play to the advantages of cloud platform in data and operation, and the traditional machine learning combined with deep learning can effectively improve the classification accuracy. Timely and accurate extraction of crop types at different spatial and temporal scales is of great significance for crops pattern change, crops yield estimation, and crops safety warning.

**Keywords:** crops classification; Sentinel-1; Sentinel-2; machine learning; deep learning; Google Earth Engine; Google Colab

## 1. Introduction

Agricultural areas in China are characterized by scattered land use types, broken agricultural landscape, and complex crops planting structures; this brings great challenges to remote sensing extraction and classification of crop types [1–3]. Currently, information

on rice fields relies on field survey data that are incomplete, time-consuming, and lacking spatial details. Crop classification based on local processing data and the traditional machine learning classification method has low efficiency and often cannot meet the accuracy requirements of complex crop classification [4]. Therefore, it is of great significance for crop extraction to require an efficient and high precision classification method [5,6].

Remote sensing technology has been widely used in crop extraction and classification all over the world. The moderate Resolution Imaging Spectroradiometer (MODIS) data with high temporal resolution and the 30 m medium spatial resolution Landsat series have been widely used in the past years [7,8]. Although they can be used for crop extraction and classification, they are suboptimal for crop plot classification in Xiangride Town. In recent years, the availability of Sentinel-1 and Sentinel-2 data has provided good data support for crop classification. They have high spatial resolution (10 m) and temporal resolution (12 days) [9,10]. Sentinel-1 is a typical SAR, which is not affected by cloud or illumination conditions. However, it has noise, making the map uncertain. On the other hand, the optical satellite image Sentinel-2 is limited by cloud cover, but it provides a vegetation index directly related to the growth stage of rice [11]. Chakhar et al. [12] confirmed that higher crop classification accuracy can be obtained by combining Sentinel-1 polarization characteristics and Sentinel-2 vegetation index characteristics. Felegari et al. [13] classified the crops by combining Sentinel-1 and Sentinel-2. The results showed that, compared with a single sensor, it increased the opportunity to check the details and obtain more reliable information. This method has been favored by many scholars to access the full range of advantages in crop classification.

With the support of such rich satellite image resources, many classification algorithms have been developed to map crop types. This can be divided into two parts: traditional machine learning classification and deep learning classification. As for traditional machine learning, support vector machine (SVM), maximum likelihood classification (MLC), random forest (RF), k-nearest neighbor (KNN), decision tree (DT), and object-oriented (OO) classification can effectively distinguish different vegetation types in remote sensing images by using unsupervised and supervised learning in the early stage of remote sensing classification [14,15]. Awad [16] used a support vector machine algorithm to classify crops from a series of Sentinel-1, Sentinel-2, A, and B images of different years. After adjusting for different required parameters such as nucleus, gamma, and cost, the accuracy of SVM was improved to more than 96% when Sentinel-1 data (VH and VV bands) were combined with Sentinel-2 images. Virnodkar et al. [17] used Sentinel-2 normalized vegetation index (NDVI) to study the potential of two more well-known machine learning (ML) classifiers, random forest and support vector machine, to identify seven categories, including sugarcane, early sugarcane, corn, water, fallow land, buildings, and bare land. The study showed that sugarcane farmland was successfully classified by Sentinel-2 NDVI time series with RF and SVM. Ponganan N et al. [18] implemented and tested super-pixel segmentation based on simple non iterative clustering (SNIC) algorithm to classify spatial clusters, and three machine learning algorithms commonly used to perform final classification (random forest, classification, and regression tree) and gradient lifting were applied to rice fields in Luang Street, Laos, part of Thung Kula Rong Hai, Thailand. Pott et al. [19] used Sentinel-2, Sentinel-1, and Space Shuttle Radar terrain mission (SRTM) digital elevation data to extract features and input them into a random forest classifier. Moran's I index and clustering K-means were used to evaluate the spatial variability of satellite features, aiming to develop a method for crop classification in Rio Grande do Sul, Brazil. The results showed that the overall accuracy of the crop classification model was 0.95. The results show that this object-oriented method is more conducive to the optimal extraction of rice fields. According to previous studies, random forest and object-oriented are considered to be excellent methods for crop mapping [17,20].

In recent years, deep learning method has set off an upsurge in the field of agricultural remote sensing recognition with the characteristics of efficient learning. Deep learning methods are used to classify crops, including deep neural network (DNN), recursive

neural network (RNN), and convolutional neural network (CNN) [21]. Deep neural network is a typical deep learning model which has been developed in crop classification scenarios. Cai et al. [22] classified crops in Champaign County, central Illinois. Based on the deep neural network model, 1322 Landsat multi temporal scenes and all six spectral bands from 2000 to 2015 were used. This shows that using temporal phenological information and using deep neural network model to train evenly distributed spatial samples in the research field can improve the classification results and training performance. Emile et al. [23] used recurrent neural network to classify rice crops in multi temporal Sentinel-1 data in Camag, France. The classification effect is good, indicating that the RNN based technology will be based on remote sensing. And it plays an important role in the classification of time series. Convolutional neural network is one of the most successful network architectures in deep learning methods [24]. In 2019, Kavita et al. [25] adopted a convolution neural network model with optimized parameter combination to classify and identify Indian pine trees. The results show that the accuracy of this method is very high. On the basis of fast RCNN, mopad combines an improved pyramid feature (RPF) module and a mixed class balance loss module. Zheng et al. [26] proposed a multi class oil palm detection method (mopad) to realize the accurate detection of oil palm trees and their growth status. In addition, for high spatial and spectral resolution images, Hu et al. [27] proposed a spectral spatial scale attention network (S³ANet). In the proposed method, each channel, each pixel, and each scale perception of the feature map are adaptively weighted to alleviate the spectral variability, spatial heterogeneity, and scale difference of crop map, respectively, to further increase the distance between different crops and reduce the misclassification effect. However, high-precision results often need to build complex training models, take a lot of time, and gain a large number of crop characteristics. It is necessary to find a simple model and filter feature in the appropriate scene.

In addition to the remote sensing data and classification algorithms developed in the past few decades, efforts have also been made to improve the classification efficiency in order to locate crop types in time [9]. The emergence of cloud-based platforms, such as Google Earth Engine (GEE), makes real-time crop monitoring and rapid and accurate crop mapping possible. For example, depending on the data integrity and convenient acquisition characteristics of the platform, GEE can timely obtain regional remote sensing data for monitoring crop growth and distribution changes, and can integrate multiple high-resolution images for precision agriculture mapping in a short time [11].

Based on the above research, the purpose of this paper is to evaluate the performance of traditional machine learning and deep learning algorithms in crop classification in the research area on the cloud platform. Based on this, a more comprehensive crop classification process is provided; on the data source, the random forest in traditional machine learning is used to screen the features extracted from Sentinel-1 and Sentinel-2 data, and the screened features are input into the deep neural network for training and prediction so as to realize the efficient and accurate mapping of crops.

## 2. Study Area and Datasets

### 2.1. Study Area

The research area is located in Xiangride Town (Figure 1), Dulan County, in the Buerhanbuda foothills southeast of Qaidam Basin, with an average altitude of 3100 m and a continental desert climate. The area is dry with minimal rain; a long, cold winter; a relatively cool and short summer; low average annual temperature; and low annual precipitation, concentrated in June to September, showing the same season of rain and heat [28]. The Xiangride River crosses the township and divides it into two parts, east and west. The river has a large annual flow and rich water resources.

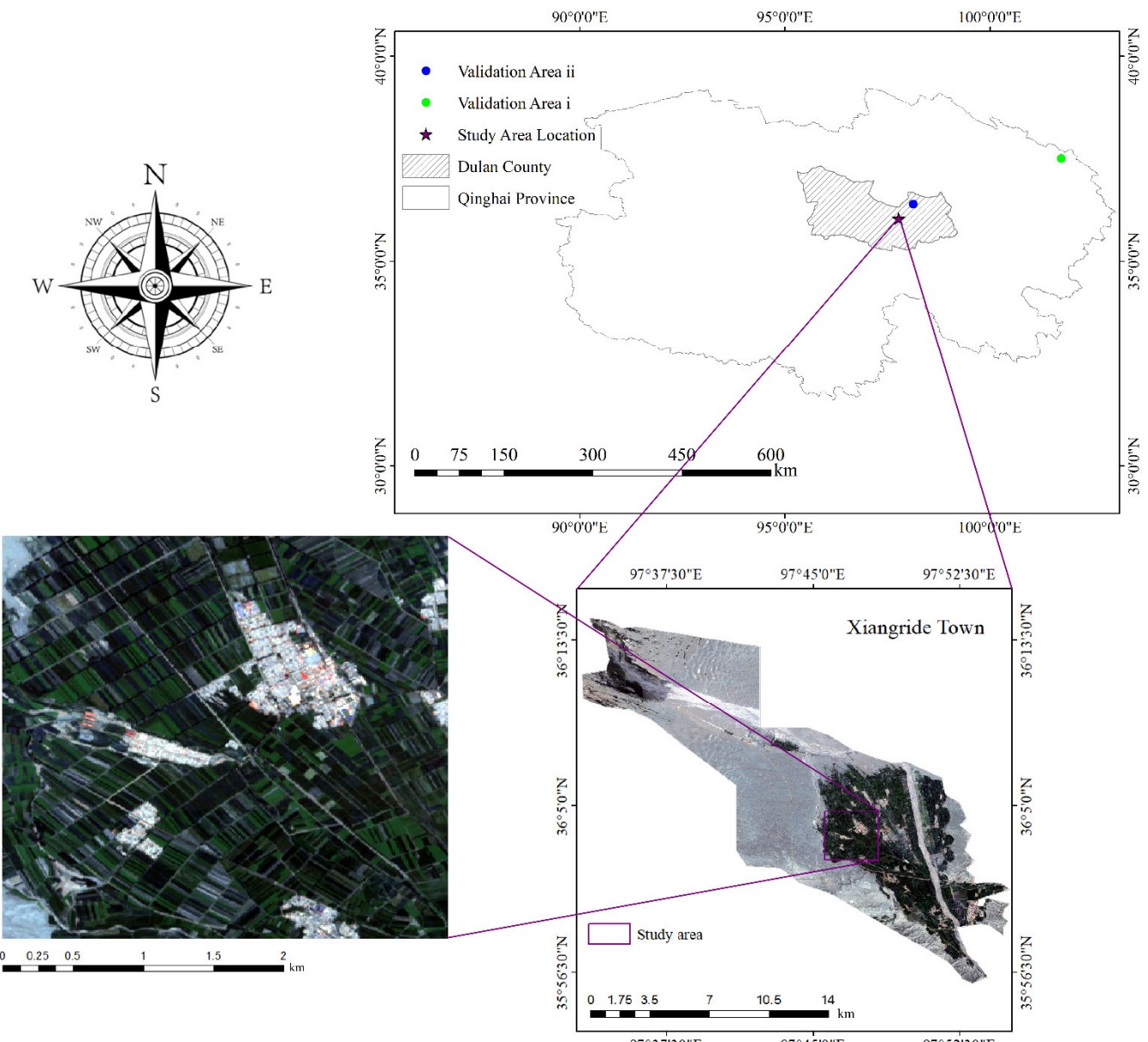

**Figure 1.** Sentinel-2 image of the study area.

Although the Xiangride town has less precipitation, it has strong solar radiation, rich light resources, and great potential of light and temperature production. The light and heat conditions are favorable to the growth of crops, and the dense river network and good natural conditions provide superior conditions for the development of irrigation agriculture [29]. The area is mainly planted with wheat, quinoa, and rape, in addition to a small part of the medlar and other cash crops. With the rapid economic development in recent years, agricultural policies and human intervention have changed the crops planting structure in this region, and it has also brought difficulties and challenges to the classification of crop types.

## 2.2. Satellite Data Processing

### 2.2.1. Data Processing Training Platform

Google Earth Engine can be accessed at https://code.earthengine.google.com/ (accessed on 10 February 2022); it is formed by Google, Carnegie Mellon university, and the United States geological survey. The platform provides more than 800 functions, and the

public data includes nearly 40 years of global satellite imagery and thematic maps, with up to 4000 images updated daily [30,31]. This greatly changes data downloading, storage, and processing, and effectively improves research efficiency and accuracy [32,33]. In recent years, GEE has been gradually used by experts and scholars in land cover classification.

Google Colaboratory (Colab) can be accessed at https://colab.research.google.com/ (accessed on 25 February 2022); it is used in machine learning research and development, as well as the depth of neural network training platform [34]. Its deep learning model training and prediction can run entirely in the cloud, which greatly relives the computing pressure of local computers. With this platform, you can write and execute code, save and share analysis results, and take advantage of powerful computing resources, all of which are free to use through the browser. In addition, the developer is provided with a Tesla T4 GPU device with about 12 GB temporary RAM and 358 GB temporary storage space, which is sufficient to meet the model training requirements in this study [34]. Therefore, this paper uses this platform for model training and prediction.

### 2.2.2. Data Processing

In this study, the Sentinel-1 synthetic aperture radar (SAR) ground range detected (GRD) and Sentinel-2 top-of-atmosphere (TOA) reflectance images (Level-1C) were used for crop type identification. Sentinel-1 and Sentinel-2 were launched by the European Space Agency (ESA). GEE has archived all Sentinel-1 and Sentinel-2 data for free use. The Sentinel-2 optical images cannot provide helpful information in cloudy and rainy weather, while the SAR satellite of Sentinel-1 can obtain usable images in cloud-covered conditions. Therefore, we used a combination of Sentinel-1 and Sentinel-2 images to find the best combination of features to classify crop types and extract crop types in the study area.

(1)　Sentinel-2 Data

Sentinel-2 is a 14-band multispectral acceptable spatial resolution optical image with 4 bands of 10 m (blue, green, red, and near-infrared), 6 bands of 20 m (red edge 1, red edge 2, red edge 3, red edge 4, swir1, and swir2), and 4 bands of 60 m (aerosol, water vapor, cirrus, and cloud cover) spatial resolution. In this study, 20 m and 60 m reflectances were resampled to 10 m by the nearest-neighbor approach [35]. Sentinel-2 consists of 2 satellites (2A and 2B) with ten days of revisit time for each satellite and an interception/overlapping period of 5 days for both satellites. Sentinel-2 TOA Level-1C images were collected from July 1 to August 31, 2021, for a total of 62 days, followed by cloud free and median synthesis. The cloud-masking band of Sentinel-2 provides information on cloud cover, used in the study for cloud free operations. The standard level 1C product ("COPERNICUS/S2") provides top of atmosphere reflectance (TOA) and is defined in Universal Transverse Mercator (UTM) map projection. Sentinel-2 TOA Level-1C data is widely used in crop type classification and has minimal impact on crop classification results [36].

(2)　Sentinel-1 Data

Sentinel-1 data comes from Google Earth Engine's primary ground distance detection (GRD) interference wide strip (IW) product ("COPERNICUS/S1_GRD") [37]. The observation results of sentry 1 radar are projected onto the standard 10 m grid output by GRD. Sentinel-1 in C band imagery (frequency = 5.4 GHz) globally had a 12- or 6-day revisit cycle depending on 1A and 1B imagery availability. The IW modes of Sentinel-1 provided dual-polarization with vertical transmit and vertical receive (VV) and vertical transmit and horizontal receive (VH). VV and VH of Sentinel-1 images with 10 m are widely used in the crop-type classification [38].

Sentinel-1 in GEE was proceeded by Sentinel-1 Toolbox using the thermal noise removal, radiometric calibration, and terrain correction. Finally, our weekly mean composited data and obtained the time series of VV and VH from 1 July to 31 August 2021, to determine the crop type for this study.

### *2.3. Samples Selection and Set*

Field surveys to investigate vegetation types and distributions in the study area were conducted from August 8 to 10, 2020, and July 17 to 24, 2021. The main sample types (Figure 2) were wheat, quinoa, rapeseed, and others; other types of land are no longer subdivided (bare land, buildings, wetlands, etc.) due to no key analysis.

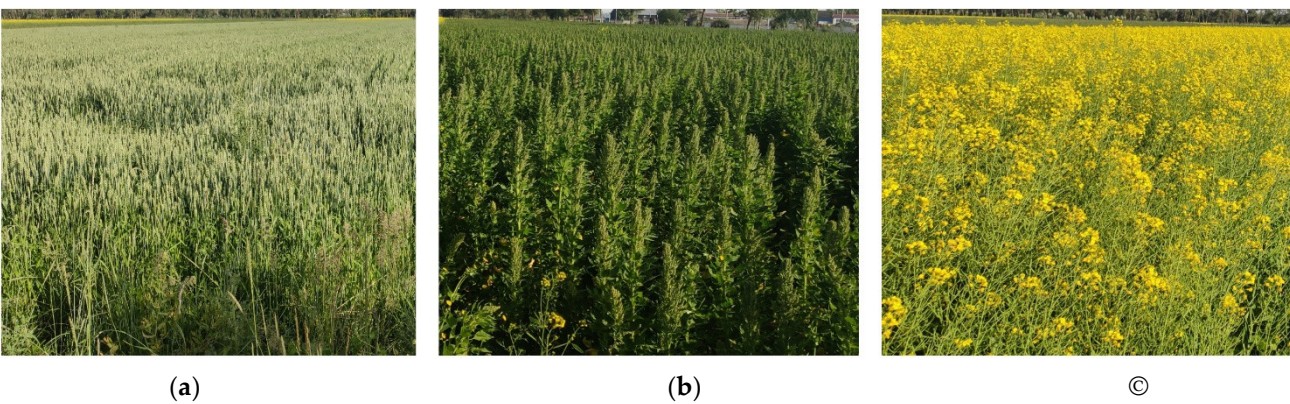

|                | (**a**) | (**b**) | © |

**Figure 2.** Field photograph: (**a**) wheat; (**b**) quinoa; and (**c**) rapeseed. Field photographs show wheat, quinoa, and rape as the three main crops studied.

To ensure the reliability and accuracy of the samples used in the study, samples were selected according to the data obtained from field investigation and the performance of different crops in remote sensing images. The specific selection and division of samples are shown in Table 1.

**Table 1.** Sample selection and division.

| Category | Type | Number | Total Number | T/V Ratio |
|----------|------|--------|--------------|-----------|
| Wheat    | G [1] | 60  |      |         |
|          | F [1] | 242 |      |         |
|          | Sum  | 302 |      |         |
| Rapeseed | G [1] | 60  |      |         |
|          | F [1] | 244 |      |         |
|          | Sum  | 304 | 1183 | 0.7:0.3 |
| Quinoa   | G [1] | 60  |      |         |
|          | F [1] | 196 |      |         |
|          | Sum  | 256 |      |         |
| Others   | G [1] | 120 |      |         |
|          | F [1] | 201 |      |         |
|          | Sum  | 321 |      |         |

[1] In the table, G refers to the actual ground survey sample points and F refers to the actual survey data selected on GEE platform. The ratio includes the number of used training sampling points (T) and validation sampling points (V) for each category.

### 3. Method

The research process is shown in Figure 3. The main contents include: (1) Data acquisition and pretreatment; (2) The preparatory work before classification mainly includes sample collection, feature analysis of normalized difference vegetation index (NDVI) time series reconstruction, and construction of feature space; (3) Crops plots were classified by random forest, object-oriented, deep neural learning, and RF+DNN using feature collaborative data; and (4) Classification accuracy and results analysis and evaluation.

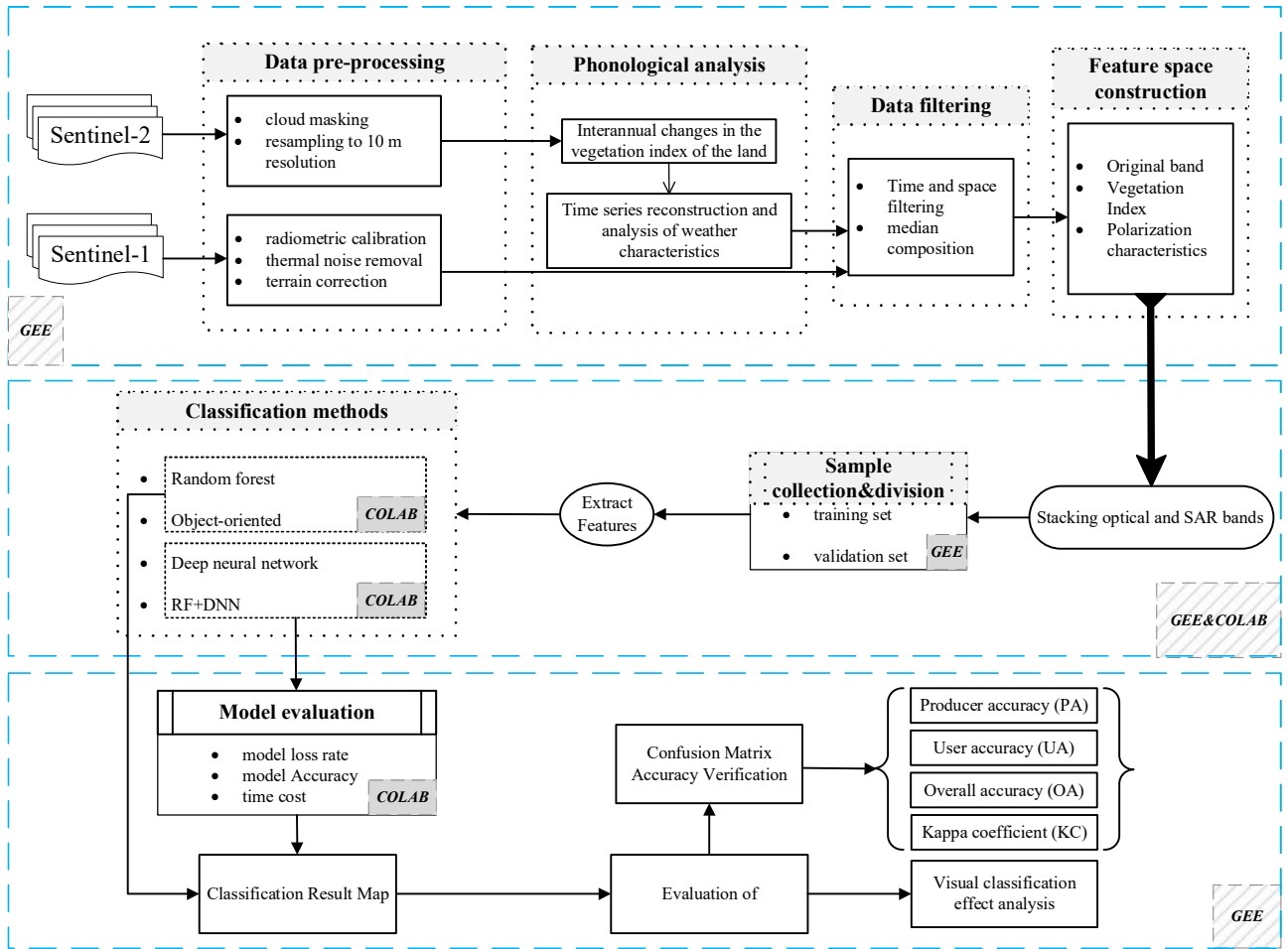

**Figure 3.** Technical route map. The GEE represents the Google Earth Engine platform and the Colab represents the Google Colaboratory platform in the flowchart.

### 3.1. Crops Timing Characteristics

Based on the Sentinel-2 NDVI sequence curve, phenological characteristics were analyzed and extracted. Based on the construction of the vegetation index time series curve, the data is rebuilt, fitted with the data by dynamic harmonic regression model (DHR) and polynomial smoothing algorithm (S-G) [39]. The radiation noise caused by sensor error and sun and atmosphere is reduced so as to extract crops growth changes and weather characteristics by curve curvature method.

NDVI show a certain regular change process with time corresponding to the physiological process of crops. Therefore, the different laws based on crops weather characteristics can be beneficial to the determination of the time range and the division of type of crops classification to some extent.

According to the sequence curve of crop growth phenology characteristics constructed by NDVI, changes occur with time in the growth cycle of crops, corresponding to the change process of increasing, reaching the peak, and decreasing, thus corresponding to the physiological process of crops from growth and development to maturity and senescence.

According to the field investigation and multi-temporal image comparison, the crops extracted in this paper have the same planting time but little difference in harvest time. Therefore, in order to reduce the influence of grassland and other vegetation on crops extraction, we extracted the long time series NDVI change curve after sowing in March according to crops sample points, and reconstructed NDVI time series using DHR and S-G algorithm (Figure 4a). The statistical results of the reconstruction effect of the two filtering methods on NDVI time series curve are shown in Table 2. The statistical parameters

use the root mean square error and correlation coefficient to reflect the closeness of the reconstructed curve to the original curve. The smaller the root mean square error, the larger the correlation coefficient and the better the fidelity of the curve. Among them, the root mean square error of DHR method is the smallest, and the correlation coefficient is the largest, at 0.1072 and 0.9158, respectively. That is to say, compared with S-G method, DHR can not only remove noise, but also better fit the original NDVI data.

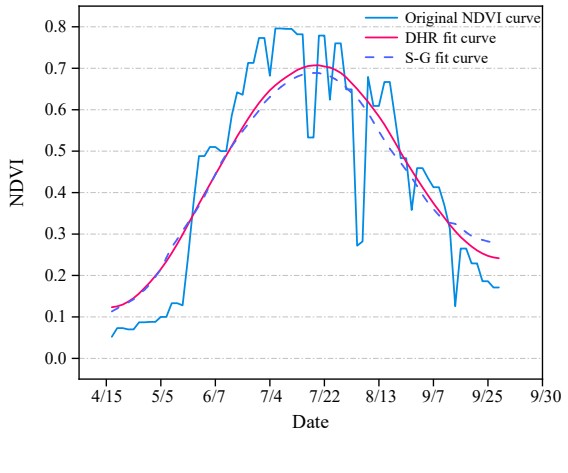

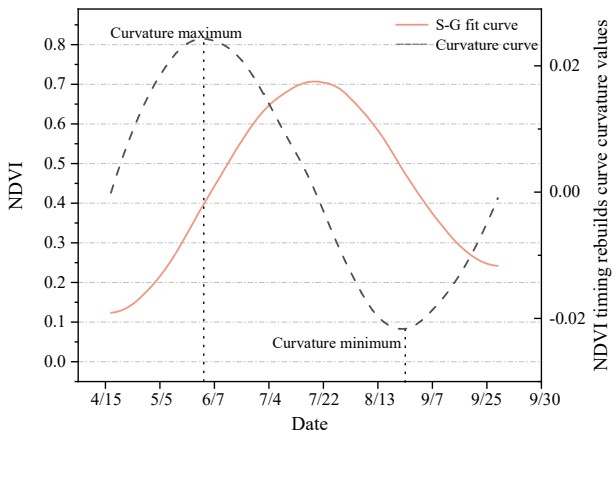

(**a**)    (**b**)

**Figure 4.** NDVI timing reconstruction and crops weather characteristic extraction. (**a**) NDVI timing refactoring, different filtering methods are used to smooth the original NDVI time series curve; (**b**) Crops weather characteristic extraction. Curvature maximum means the location with the fastest NDVI improvement, and curvature minimum means the location where NDVI decreases the fastest.

**Table 2.** NDVI timing refactoring parameter statistics.

| Index | Raw Data | HDR Data | S-G Data |
|---|---|---|---|
| Mean | 0.4389 | 0.4458 | 0.4389 |
| Mean square root error | | 0.1072 | 0.1135 |
| Correlation coefficient | | 0.9158 | 0.9109 |

Based on the NDVI time series curve reconstructed by DHR, the curve curvature method was used to analyze the phenological characteristics of crops so as to analyze the different stages of crop growth (Figure 4b). According to the maximum value of curvature and minimum value of curvature in the figure, we can analyze that the fastest growth period of crops is in mid-June, while the maximum aging time of crops is in early September. Rising curvature represents crops growth, and vice versa. Based on the field knowledge of local climate factors and other influences, local crops mature faster. Therefore, the study time dimension is from 1 July to 31 August 2021, when crops grow faster, based on the multi-factor analysis of phenological characteristics and image comparison.



*3.2. Feature Space Construction*

One of the key steps in remote sensing image classification is to construct classification feature space, and the selection of classification feature is particularly important. In this paper, Sentinel-1 and Sentinel-2 data are processed and selected accurately. The original band, vegetation index feature, and polarization feature were selected to construct feature space for superposition classification analysis.

### 3.2.1. Original Band and Vegetation Index Feature

In many studies on vegetation change and classification, most of them focus on the relationship between vegetation and vegetation index (VIs), among which the NDVI is the most common and most commonly used vegetation index. The Sentinel-2 data used in this paper has high spatial resolution and multiple red-edge bands [40]. Therefore, the original 13 bands and four kinds of normalized differential vegetation indices are extracted on this basis: NDVI, NDVIre1, NDVIre2, and NDVIre3, and their calculation is shown in Table 3.

**Table 3.** Vegetation index used in the article. The bands in the calculation formula correspond to the Sentinel-2 bands.

| Vegetation Index | Formula (Sentinel-2) |
|---|---|
| NDVI | $(B8\,[1] - B4\,[1])/(B8\,[1] + B4\,[1])$ |
| NDVIre1 | $(B8A\,[1] - B5\,[1])/(B8A\,[1] + B5\,[1])$ |
| NDVIre2 | $(B8A\,[1] - B6\,[1])/(B8A\,[1] + B6\,[1])$ |
| NDVIre3 | $(B8A\,[1] - B7\,[1])/(B8A\,[1] + B7\,[1])$ |

[1] The B in the table represents the image band of Sentinel-2 satellite. The B8A refers to Narrow NIR, whose central wavelength is 0.842 μm. The B4 refers to Red, whose central wavelength is 0.665 μm. B5, 6, and 7 are three red-edge bands of vegetation, and their central wavelengths are 0.705 μm, 0.740 μm, and 0.783 μm, respectively.

### 3.2.2. Polarization Feature

Sentinel-1 radar data, a high-resolution imaging system, is free of cloud and rain and can monitor the surface all day and all night. In this paper, multi-temporal Sentinel-1 radar images are selected to discuss the influence of multi-temporal radar remote sensing data on the accuracy of ground object classification [41].

The time series images of VV polarization and VH polarization in the study area during the period of good crops growth determined by time series analysis were obtained on the GEE platform, and the backscattering coefficients of the sample points of various object types were calculated. The mean value operation was carried out according to the time period of weeks, and the variation rule of backscattering coefficients of different crops with time was studied.

*3.3. Classification Method*

In order to highlight the advantages of the cloud platform processing image and classification process, this paper adopts pixel-based random forest classification, sample-based object-oriented classification, deep neural network classification, and a classification method combining random forest and deep neural network. The classification process of random forest, object-oriented, and deep neural network is shown in Figure 5.

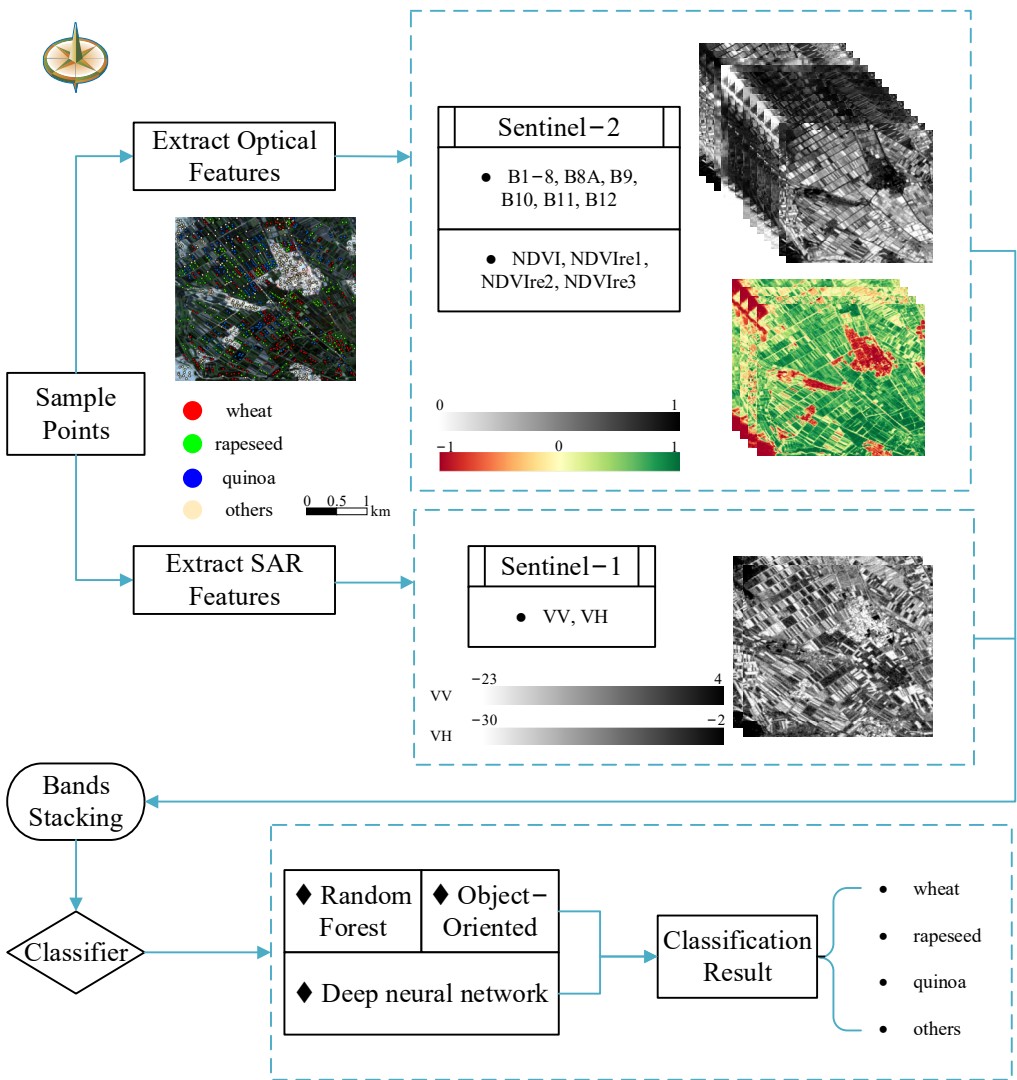

**Figure 5.** Technical route map. The GEE represents the Google Earth Engine platform and the Colab represents the Google Colaboratory platform in the flowchart.

### 3.3.1. Traditional Machine Learning

Random forest and object-oriented classification are traditional ground object classification methods based on pixels and objects. In order to compare the advantages and disadvantages of the two algorithms in the process of ground object extraction, the feature space including original band, four kinds of normalized difference vegetation index, and polarization feature is constructed in this paper. Sample data were divided into training sample set and verification sample set in a ratio of 0.7:0.3, and the full feature space was input into random forest classifier to classify wheat, quinoa, rape, and others. In the object-oriented classification method, SNIC algorithm is used to initially segment the image, and after the relevant adjustment and optimization of parameters to achieve relatively optimal segmentation effect, classify the different categories of crops according to the segmented object.

(1)    Random Forest Classification

The learning classifier of the decision tree-based series integrated machine proposed by Leo Breiman is based on the principle of using the Bootstrap resampling method to randomly re-sample about two-thirds of the samples from the training sample and repeat it several times, and to generate the decision tree for each training sample, leaving about one-third of the training samples as out-of-bag data for internal cross-testing to assess the classification accuracy of random forests. The GINI coefficient is used to determine the

split conditions of each node in the decision tree, and the construction of random forest is completed [42].

According to previous studies, good results can be obtained by using 60 trees, and higher values will not significantly improve the accuracy of classification [17]. In addition, previous studies have shown that the RF method is not very sensitive to the number of features randomly selected to segment each node [18]. Therefore, the Ntree was set as 60 and the default parameter of Mtry was utilized in our study.

(2)  Object-Oriented Classification

Object-oriented classification technology sets near cells as objects to identify spectral features of interest, making full use of high-resolution full-color and multispectral data space, texture, and spectral information to segment and classify the characteristics of high-precision classification results or vector output [43]. GMeans algorithm, KMeans algorithm, and SNIC algorithm are provided for the image segmentation GEE platform. The image segmentation results are better than the first two as the SNIC algorithm has been adjusted and optimized by the parameters, so this algorithm is used for image segmentation [44].

In the process of using SNIC for object-oriented crop classification, some parameters need to be set. After many experiments, when compactness is set to 0.1, connectivity is set to 8, neighborhood size is set to 256, and seeds is set to 13, the segmented unit is more suitable for the plot scale of crops.

### 3.3.2. Deep Learning

The deep learning models used in this experiment are all built under the TensorFlow framework, version 2.8.0. The experiment is conducted on NVIDIA Tesla T4, and the CUDA and CUDNN versions are 11.2 and 8.1, respectively. When we select the optimization function, we fully consider the problems of time, model cost, and performance. As the adaptive moment estimation method makes each parameter obtain an adaptive learning rate in the process of model training optimization, we may achieve the double improvement of optimization quality and speed. Therefore, the optimization is guided by setting the adaptive moment estimation (Adam) optimizer. During the experiment, different network structures are compared, such as adjusting the number of network layers and the number of neurons in each layer. In addition, the batch number is also compared and analyzed. Finally, it is determined that when three hidden layers are used and the number of neurons is 64, 32, and 16, respectively, the model training and prediction effect is the best.

(1)  Deep Neural Network Classification

Deep neural network refers specifically to fully connected neuronal structures that do not contain convolution units or time associations and are based on multi-layered sensor MLP (or artificial neural network ANN) [21]. Its internal neural network layer can be divided into three categories, input layer, hidden layer, and output layer. The input layer of this classification method includes all the features in the built feature space, and three implicit layers are designed to predict the classified image. In this model, after many tests, three hidden layers are set, and the neurons of the three hidden layers are 64, 32, and 16 in turn. Dropout is set to 0.2 and batch to 64, and a total of 5000 epochs are used to train the model.

(2)  RF+DNN Classification

In the process of classification, the characteristics that affect the classification of land objects often have many and different effects, which not only produce data redundancy but also may lead to the occurrence of false ground material [45]. Therefore, this paper designs a crop block classification framework of first feature screening and then prediction classification.

During the calculation of GINI, the Ntree was set as 60 and the default parameter of Mtry was utilized in our study. Based on the characteristics of importance as a result, we studied the influence of the characteristic quantity of classification accuracy according to the computing research GINI index classification accuracy with the characteristics of the importance of variation (Figure 6a). When the characteristic number increased from 1 to 3, classification accuracy showed a trend of rising faster. According to the classification overall accuracy and Kappa coefficient change trend graph analysis, we determined the threshold of feature selection; this paper adopts a method of curvature changes comparison to determine the characteristics of the number of threshold (Figure 6b). And the Kappa coefficient change with the characteristics of the line chart of Loess smoothness, after the first-order linear differential smooth curve. According to the variation range of differential curves with features, the features were selected for classification, that is, when the number of features increased to 12, the change rate curve of Kappa curve basically stabilized, i.e., the change of Kappa coefficient basically remained gentle.

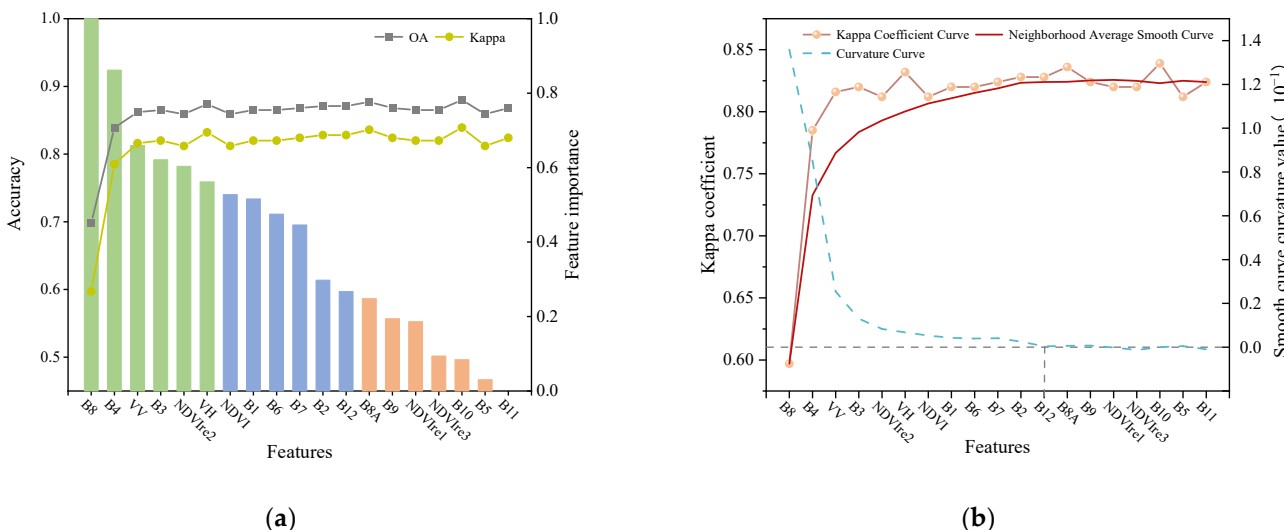

(**a**)    (**b**)

**Figure 6.** The relationship between features and classification accuracy and the graph of feature screening methods. (**a**) Variation of classification accuracy with features; (**b**) Determine feature threshold diagram.

The platform is used to process images and features, input features into random forest classifier, use the Gini index as an evaluation index to sort the importance of feature space features, filter features according to the importance and overall accuracy of feature size and the relationship between Kappa coefficients, convert them into a Colab recognizable data format (TFRecord), and input depth neural network model for training and prediction. Finally, the prediction results and accuracy evaluation are processed and displayed on GEE platform (Figure 7). The only difference between this experimental content and the DNN model is the difference in input data. The input characteristics are filtered by GINI value first. The model structure and parameter settings are the same as those of the DNN model. In this model, we also set up three hidden layers in the DNN network model. The neurons of the three hidden layers are 64, 32, and 16 in turn. Dropout is set to 0.2, the batch is set to 64, and a total of 5000 epochs are used to train the model.

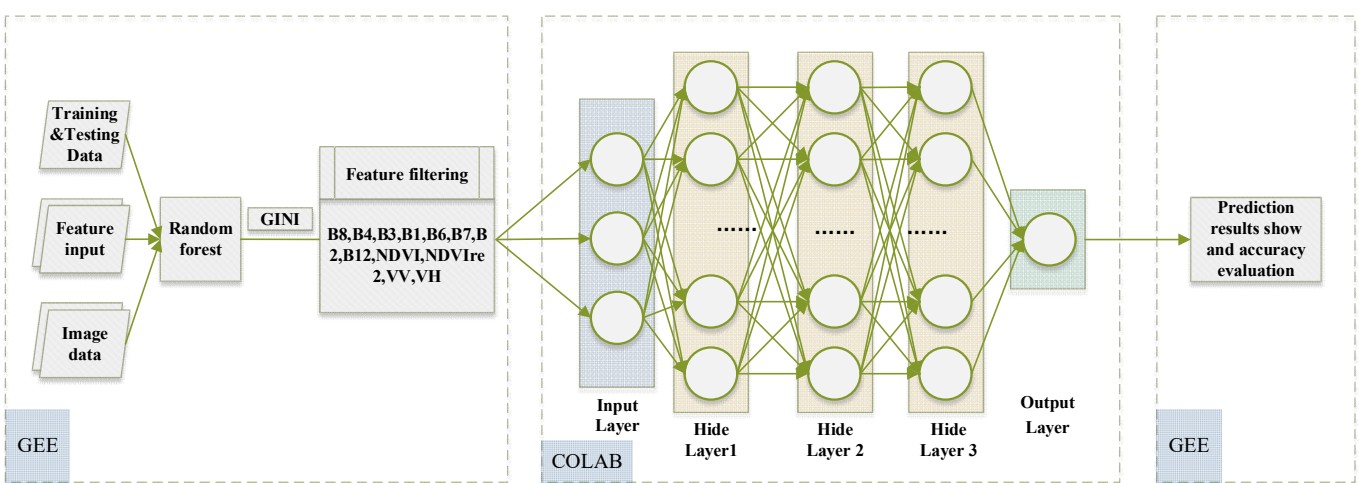

**Figure 7.** RF+DNN classification method flowchart. This flow chart illustrates the overall framework of the RF+DNN classification method proposed in this study.

*3.4. Classification Features and Accuracy Assessment*

3.4.1. Assessment of Features Importance

Ideally, more variables would lead to a better representation of the features of LC and improve the mapping accuracy. However, previous studies have shown that the accuracy and efficiency of the classification is affected by the high correlation of the data, the noise in the data collection and correction process, and the increase in computational complexity [46].

As a measurement for node impurity, the value of GINI can indicate how often a random instance will be misclassified. Therefore, GINI can be used to evaluate the importance of each variable [47]. In the RF model, GINI is calculated over all trees as the averaged reduction in node impurity on one splitting variable. As significant variables can substantially reduce the impurity of the sample, the higher GINI values represent greater importance of the variable. This index was used as a criterion to evaluate the importance of the variables [48]. Technically, the GINI index can be calculated in GEE directly.

3.4.2. Accuracy Assessment

The classification result is evaluated by confusion matrix method. It is a standard format for classification accuracy evaluation. It is a comparison array used to represent the number of real pixels divided into a certain category and the number of pixels tested in the experiment, which is represented by a matrix of N rows and N columns. These indicators include producer accuracy (PA), which refers to the ratio of the number of pixels correctly divided into this category and the reference total number of real pixels of this category. User accuracy (UA) refers to the ratio of the total number of pixels correctly divided into this category to the total number of pixels divided into this category. Overall classification accuracy (OA) is the overall evaluation of the quality of classification results. Kappa coefficient (KC) is an index used to test whether the predicted results of the model are consistent with the actual classification results and can be used to measure the classification results [15]. The model accuracy evaluation of deep neural network and RF+DNN is evaluated from the model loss rate, model accuracy, model training time, and prediction time [34].

**4. Results**

*4.1. Feature Space Analysis*

The spectral characteristics of different land features include 13 bands of Sentinel-2 multispectral data and four vegetation indices calculated based on them, including normalized differential vegetation index and three NDVI combined with red-edge bands. It

can be seen from the spectral characteristic heat map (Figure 8a) and characteristic importance histogram (Figure 8b) of crops that most spectral features can distinguish crops from other land types, but only a few original bands have good discrimination between crops, and only NDVI and NDVIre2 play a comparative role in the four added vegetation indexes. Therefore, only relying on spectral characteristics is not enough to distinguish crops well.

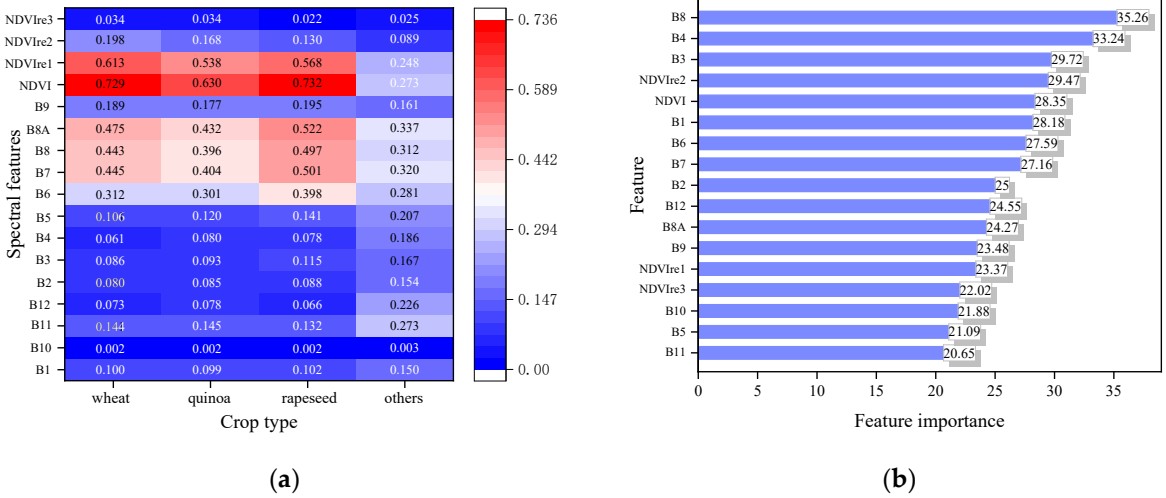

(**a**)　　　　　　　　　　　　　　　　　(**b**)

**Figure 8.** Crop spectral characteristic heat map and feature importance histogram. (**a**) Spectral characteristic heat map of different crop type; (**b**) According to the statistics of the characteristic scores output by the random forest model.

The polarization characteristic is related to such factors as the crops growing conditions; the knowledge during the study period according to the polarization characteristics of radar data using Sentinel-1 can better distinguish between different features, which can be an important basis for the classification of different features. The polarization characteristics of different land features are further obtained by using Sentinel-1 data. The calculation of polarization characteristics takes the starting time as the starting point and divides the time period into weeks. The mean value of each week is calculated to obtain the mean curve of backscattering coefficients of different ground objects (Figure 9).

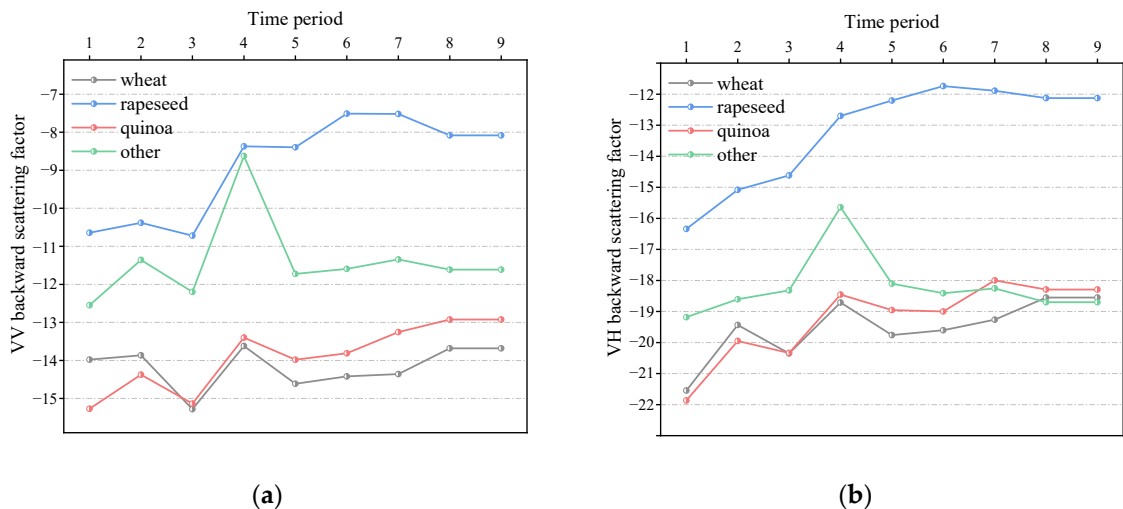

(**a**)　　　　　　　　　　　　　　　　　(**b**)

**Figure 9.** Polarization characteristic curves of different features. (**a**) VV polarization feature curve; (**b**) VH polarization feature curve.

*4.2. Analysis of Classification Results*

In this study, the use of a cloud platform simplifies image preprocessing and improves the classification efficiency. However, the effect of using traditional machine learning methods to classify crops is not ideal. In order to further improve the classification accuracy, DNN and RF+DNN classification methods are introduced to classify crops.

Based on GEE and Colab cloud platforms, DNN is used to train and predict the image by using selected ground object sample points and inputting all features. The model loss rate was 0.2957, the model accuracy was 0.8674, and the total time spent on model training was 36 min 11 s. After that, the trained model was used to classify the image plots in the study area, and the total time was 58 s (Table 4). For RF+DNN model, the model loss rate was 0.2386, the model accuracy was 0.9042, and the total time spent on model training was 17 min 23 s when using filtering features. The trained model was used to classify the types of image plots in the study area, and the total time was 11 s. Compared with the DNN classification model, the RF+DNN classification model has smaller loss rate, higher accuracy, and shorter train and predicted time consumption. In addition, compared with the traditional classification methods, namely random forest and object-oriented classification, this classification method has better classification accuracy and effect.

**Table 4.** Comparison of DNN and RF+DNN classification methods.

| Type | Model Loss | Model Accuracy | Training Time | Prediction Time |
|---|---|---|---|---|
| DNN [1] | 0.2957 | 0.8674 | 36 min 11 s | 58 s |
| RF+DNN [1] | 0.2386 | 0.9042 | 17 min 23 s | 11 s |

[1] The hardware support of the computing process is carried out using the GPU provided by the Colab platform.

In order to further highlight the classification effects of various classification methods on different ground objects, the following will be analyzed and explained from the classification accuracy and classification result image.

Firstly, the producer precision and user precision of classification results are analyzed, and different classification methods are evaluated according to these data. The classification algorithms used in this study include RF, OO, DNN, and RF+DNN classification method. Figure 10a shows the PA and UA of three crops calculated for four different classification schemes using synthetic aperture radar and optical images. It shows the distribution of user accuracy and producer accuracy of the four classification methods. According to their performance, the extraction accuracy of rape crops performs better in the four methods, because rape crops are convenient for other crops from the perspective of spectrum. However, the accuracy of wheat and quinoa crops is lower than that of rape crops, which is due to their similar performance in remote sensing images, and the appearance of wheat and quinoa crops observed in the field is also relatively similar.

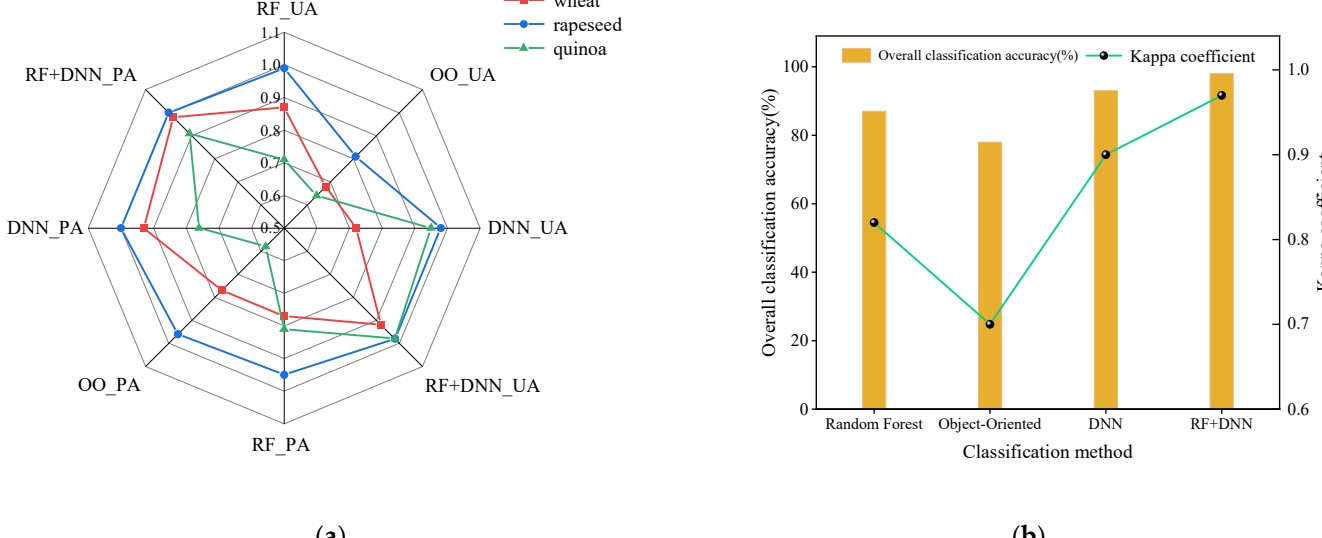

**Figure 10.** The map of the accuracy of different classification methods. (**a**) Producer accuracy and user accuracy graphs of different crops under different classification methods (RF, OO, DNN, and RF+DNN). (**b**) The overall accuracy and Kappa coefficient of different crops under different classification methods.

According to the classification accuracy distribution diagram of a single crop, the effect of object-oriented classification is relatively poor. Except for rapeseed, the extraction accuracy of wheat and quinoa is relatively low. Compared with object-oriented classification, random forest classifier can improve the extraction accuracy of wheat and rape crops. However, wheat and quinoa crops still have misgrades and omissions. In order to reduce misclassification, DNN was used to classify the three crops. The results showed that the overall accuracy was improved, but the improvement effect was not significant for wheat and quinoa crops. RF+DNN classification not only has the highest classification accuracy on the whole, with the classification accuracy of individual crops reaching more than 90%, but also greatly inhibits the misclassification and omission phenomenon of individual crops.

Four classification methods are analyzed from the perspective of overall accuracy and Kappa coefficient (Figure 10b). Among them, the overall accuracy of the RF method is 87% and the Kappa coefficient is 0.82, while the overall accuracy and Kappa coefficient of OO classification are 78% and 0.70, respectively. The classification accuracy of random forest classification is higher than that of object-oriented classification. The results of crop extraction by DNN and RF+DNN classification methods have been greatly improved (OA = 93%, Kappa = 0.90) and the accuracy has been improved. The accuracy of crop classification using RF+DNN is the highest (OA = 98%, Kappa = 0.97).

According to the overall classification results in Figure 11, the distribution of the three crops in the study area is relatively mixed and lumpy, and some roads with linear distribution between the plots are also obvious. As for the performance of different ground objects in the results of different classification methods, they are excellent in the extraction of rape, wheat, and quinoa. Compared with the other three classification methods, the object-oriented classification method has a more obvious edge extraction. However, it is not friendly to some small plots, and its extraction of quinoa and random forest classification method are poor. They tend to mistakenly classify quinoa into other land groups. In contrast, the extraction of DNN and RF+DNN of these three crops is much better. DNN classification results show that the extraction effect of quinoa is much better than RF and OO, but the accuracy of wheat will be reduced due to classification errors. In general, the RF+DNN classification method not only greatly improves the crop extraction accuracy, classification efficiency, and model stability, but also highlights the distribution

of spatial features of ground objects. It is not only the best ground object classification method set in this research field, but also a reference for ground object classification based on machine learning combined with deep learning in the future.

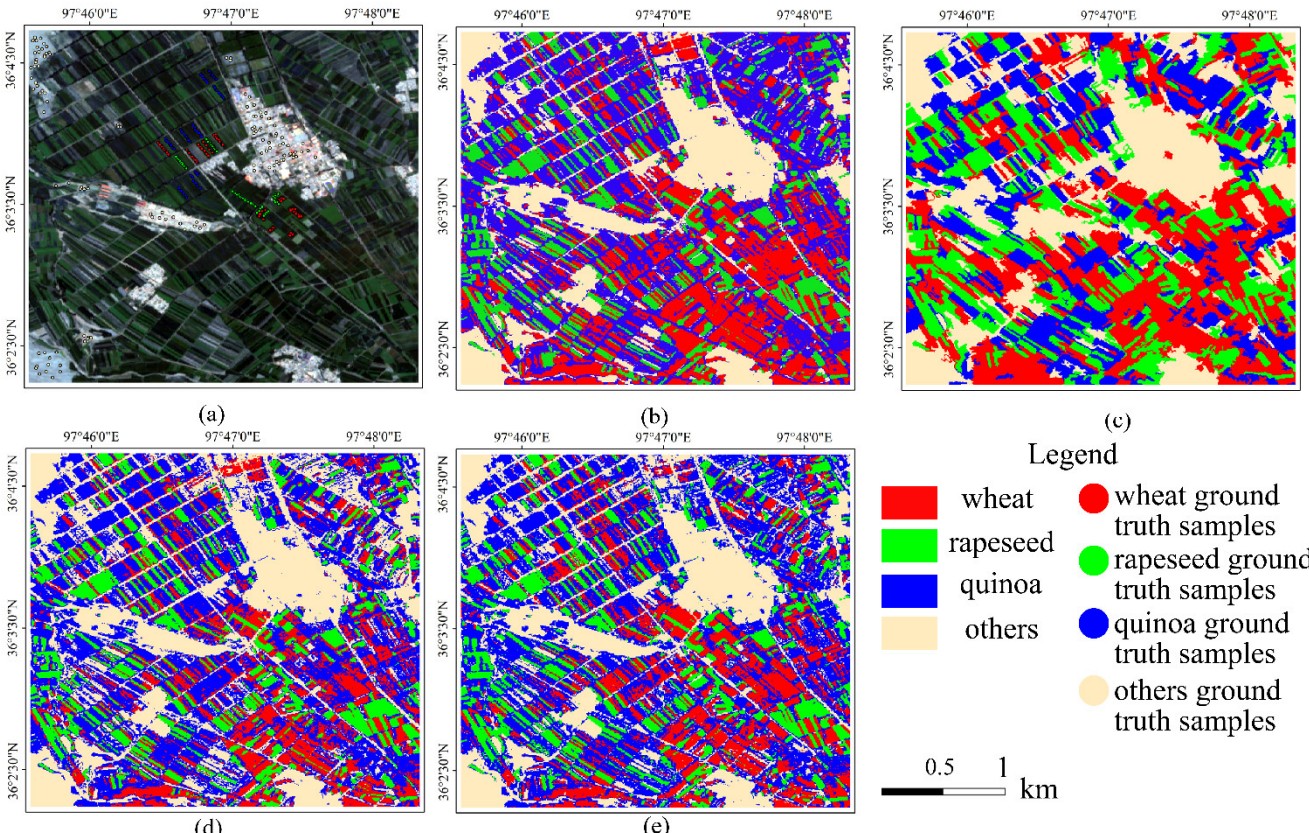

**Figure 11.** Result diagram of different methods classification. (**a**) Study area true color image (Red, Green, Blue) display; (**b**) Classification results of random forest classification method; (**c**) Classification results of object-oriented classification method; (**d**) Classification results of deep neural network classification method; and (**e**) Classification results of RF+DNN classification method.

## 5. Discussion

### 5.1. Model Transfer and Validation

In this section, we discuss the transferability and generalization capability of the proposed RF+DNN model. To this end, we selected two validation areas (Figure 1: Validation Area i, Validation Area ii) for analysis [49,50]. The purpose is to use Validation Area i to verify the model training and prediction effect in the new area and use Validation Area ii to verify the performance of model transfer, so as to reduce the contingency and improve the universality of the results. In full consideration of phenology and crop type distribution, we first selected the location far away from the study area of this paper (Validation Area i) for research. The main purpose of the study in this area is to verify whether there is a good classification effect by using the same network structure in a large area far from the original study area. In order to further evaluate the performance of the model, we have introduced precision, recall and F1-score to evaluate the quality of the model [51]. Precision describes the ability of the model to correctly detect crops, while recall describes the ability of the model to detect real crops, and F1-score evaluates the overall performance of the model. They can be calculated by the following formula:

$$\text{precision} = \frac{\text{TP}}{\text{TP} + \text{FP}} \tag{1}$$

$$\text{recall} = \frac{TP}{TP + FN} \tag{2}$$

$$F1 - \text{score} = \frac{2 \times \text{precision} \times \text{recall}}{\text{precision} + \text{recall}} \tag{3}$$

where TP means true positives, FP means false positives, and FN means false negatives.

In this area, a total of 2386 sample points are selected (Figure 12a). Similarly, the features screened in Figure 12 are input into the network structure. The network structure contains three hidden layers, with 64, 32, and 16 neurons, respectively. Dropout is set to 0.2, and Adam optimization function is used for training and prediction, the relevant parameters are the same as those set by RF+DNN. The results showed that the precision was 0.9378, recall was 0.9299, and F1-score was 0.9338. According to the evaluation index results of these three models, the value distribution is above 0.9. In addition, the value of the model loss is 0.1947, and the accuracy also reaches 0.9341 (Table 5), which fully shows the superiority of this model.

On the basis of the above, we further use model transfer to verify the generalization ability of the model, which will be displayed in Validation Area i migrating the trained model to Validation Area ii, extracting the crops in the area. In Validation Area ii. 873 crop samples, extracted features, data set setting and Validation Area i consistent. The results show that the loss of the model is 0.0976 and the accuracy reaches 0.9812. The model transfer has good performance.

**Table 5.** Validation Area Model and Classification Accuracy.

| Validation Area | Model Loss | Model Accuracy | OA [1] | KC [1] |
|---|---|---|---|---|
| Validation Area i | 0.1947 | 0.9341 | 0.95 | 0.93 |
| Validation Area ii | 0.0976 | 0.9812 | 0.98 | 0.98 |

[1] The OA refers to the overall accuracy and KC refers to the Kappa coefficient.

For the classification accuracy and classification result map in the two areas, the overall classification accuracy in Validation Area i is 0.95, and the Kappa value is 0.93, while the overall classification accuracy in Validation Area ii is 0.98, and the Kappa value is 0.98 (Table 5). For the classification result map (Figure 12), the two classification results can better extract the crop plots. Through the verification of the model in these two verification areas, it can be judged that the model has good performance, transferability, and generalization ability from the qualitative and quantitative point of view.

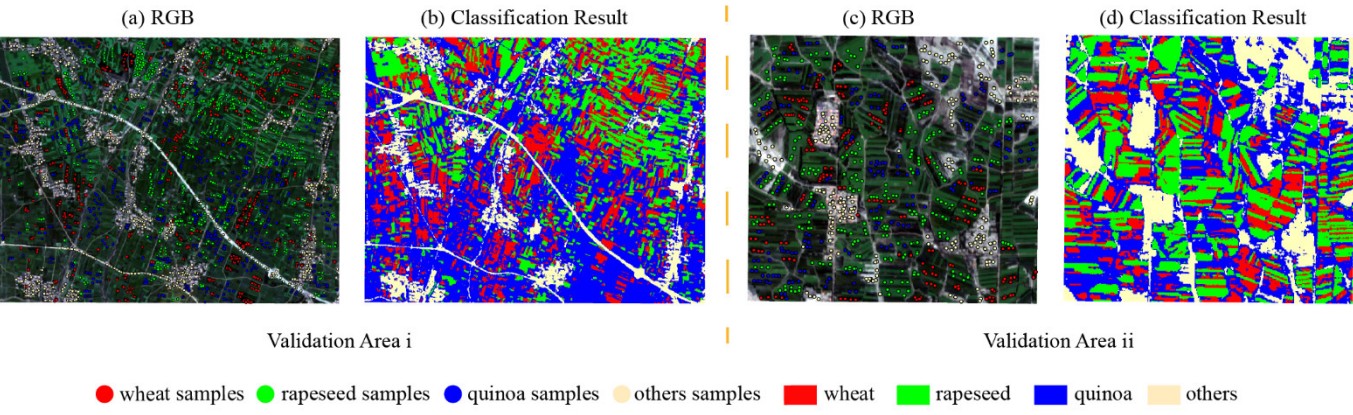

**Figure 12.** Validation area classification. (**a**) Validation Area i true color image (Red, Green, Blue) display; (**b**) Classification results of Validation Area i; (**c**) Validation Area ii true color image (Red, Green, Blue) display; (**d**) Classification results of Validation Area ii.

*5.2. Feature Contributions to Crops Classification*

In this paper, different classification methods are compared, and on this basis, the RF+DNN classification method is introduced. The first task is to participate in the classification of crops characteristics screening. The feature space constructed in this paper includes original band, vegetation index, and polarization characteristics, which plays a huge role in the subsequent analysis and research.

The original band is the most essential manifestation of ground objects in remote sensing image [12]. The difference of band between different ground objects can effectively classify crop types to a certain extent [9]. In this paper, the red-edge vegetation index, which is sensitive to vegetation, is introduced based on the band, which further expands the difference on the basis of the difference of the original band and is more conducive to the improvement of the accuracy of ground object classification. In addition, phenological analysis of the study area was carried out according to vegetation index, and images of appropriate time phases were selected for synthetic classification. However, the original band and red edge vegetation index are the spectral features of ground objects, and their mechanism is relatively singly focused, which is susceptible to the same factors, leading to poor classification [35]. Therefore, the polarization feature is introduced to reduce the influence of external factors on classification effect.

The combination of optical and radar images can make up for the influence of optics susceptible to cloud and snow and radar data noise in a certain situation. However, if the texture of some crops is well developed, the texture feature is often a relatively important content, which is easier to reflect the surface properties of different crops in the image [19]. Therefore, texture features will be added into the process of crop classification in the following study of crop classification.

*5.3. Advantages of Remote Sensing Cloud Platform in Crop Classification*

In this study, two remote sensing computing cloud platforms were used to obtain the experimental results. One method was to use Google Earth Engine to prepare, screen, and preprocess data sources required by the research, and calculate relevant indices and conduct vegetation phenology analysis according to the obtained data. In addition, this platform was also used to process the classified data and evaluate its accuracy. This method not only saves a lot of download time and storage space compared with the traditional local, but also is far more efficient in processing and analysis [10,11,15]. In addition, this experiment realized DNN and RF+DNN classification methods using Google Colab deep learning training and prediction cloud platform, and achieved good classification results.

In the development of modern agriculture, precision agriculture and real-time crop detection are becoming more and more important. The emergence of Google Earth Engine and Google Colab provides a good platform for precision agriculture and real-time crop detection. Precision agriculture often depends more on high-quality data sources and excellent crop classification and extraction algorithms [30,31]. The GEE platform provides a large number of remote sensing data sources with high and medium spatial resolution, uses sentinel series data and Landsat data, and provides corresponding preprocessing algorithms. the more mature deep learning algorithm also has a good performance in crop classification, so using the GEE and Colab platforms can realize more accurate crop classification and extraction so as to meet the needs of precision agriculture [12,13,15,16].

On the other hand, real-time monitoring of crops includes crop quality monitoring, pest monitoring, and real-time monitoring of crop changes [19,22,24]. They rely on remote sensing images with high time resolution and have high requirements for the timely acquisition of data sources. Local processing often requires a large number of remote sensing data for crop monitoring, and there will often be a certain lag effect. It is more convenient to use the cloud platform for real-time crop monitoring. It can quickly and efficiently call the latest images and batch process the long-time data so as to more easily ascertain

the detection results of crops in a certain time range. Remote sensing cloud platform and cloud computing will definitely become a hot topic in the future.

### 5.4. Significance of RF+DNN for Crops Classification

The accuracy of crop type mapping is mainly affected by the frequency and quality of remote sensing images as well as the quantity and representativeness of training data and classification characteristics [46]. Multi-sensor synthesis improves data availability and classification accuracy of crop types. For different imaging methods, data sources complement each other and play their respective advantages in crop extraction [24,35]. Multi-temporal remote sensing images can improve crop classification accuracy. Although Landsat data have been applied to some agricultural research, a spatial resolution of 30 m is insufficient in most smallholder farming systems [35]. Lidar data or high spatial resolution imagery is also recommended to improve crop type mapping in smallholder areas. The combination of Sentinel-1 and Sentinel-2 images provides more observations, and more importantly, classes with similar spectral properties have distinguishable features [4,11].

In this study, although using optical and radar data to construct feature space solves the influence of the outside world on the accuracy of crop classification to a certain extent, it also increases the computational cost. Therefore, we processed the feature space then formed the optimal classification method of RF+DNN. The RF+DNN classification method has good scalability, which can effectively compress the classification features and reduce the time cost. At the same time, it also effectively improves the classification accuracy in the study area. It has important application significance for the classification and extraction of crops or different types of vegetation.

### 5.5. Uncertainty of RF+DNN Classification Method

Although the RF+DNN classification method improves the accuracy of crop classification in the study area to a certain extent, and saves time and cost, it also has a weakness of in that the performance of crops in remote sensing images is complex and has various characteristics. At present, there are many deep neural network structures such as DNN, CNN, RNN, and other extended network structures [21,22,24,27]. Although the selectivity of the deep neural network used has increased, for different research areas the choice of what kind of network structure often requires many tests, which makes the spatial scalability of RF+DNN weak. However, the crop classification framework of RF+DNN is complete, and the uncertainty of the network model also reflects that this classification framework has good scalability. In future research, we will further find a more stable and excellent crop classification framework to realize the accurate classification of crops.

## 6. Conclusions

The production of detailed crop classification maps is essential for monitoring natural resources and tracking agricultural sustainable development goals. Based on the remote sensing cloud platform, this paper proposes a classification framework that uses the random forest model to screen the features and input them into the depth neural network model so as to realize the accurate mapping of crops. Combined with Sentinel-1 and Sentinel-2 data sources, the phenological characteristics of vegetation in the study area are analyzed, and the optical images and radar data in an appropriate time period are selected. The feature space is constructed by extracting strip features, four normalized difference vegetation index features, and polarization features. By comparing different crop classification methods, the feasibility and potential of RF+DNN combined classification method are analyzed and verified. Compared with other classification methods, the classification accuracy of RF+DNN classification method is higher (OA = 0.98, Kappa = 0.97), and the classification results are more consistent with the crop distribution in the study area. The accuracy and efficiency of model training and prediction are also better than

DNN classification method. The results show that the cloud platform based on GEE and Colab can realize the rapid acquisition and joint use of data sources and can achieve more efficient data preprocessing. The use of a cloud platform and the organic combination of traditional machine learning and deep learning can realize precision agriculture and real-time crop monitoring.

**Author Contributions:** Conceptualization, J.Y. and. Z.Z.; methodology, J.Y., Z.Z. and J.L.; software, J.Y.; validation, J.Y., C.X. and Z.Z.; investigation, J.Y., C.X. and Z.Z.; writing—original draft preparation, J.Y. and J.W.; and writing—review and editing, J.Y., J.W. and Z.Z. All authors have read and agreed to the published version of the manuscript.

**Funding:** This work was supported by the open project of key laboratory of geological processes and mineral resources in Northern Tibet Plateau of Qinghai Province under Grant 2019-KZ-01 and the special project for innovation platform construction of science and technology department of Qinghai Province under Grant 2019-ZJ-T04.

**Data Availability Statement:** The code used in this article is public and can be found on https://github.com/RGISYJX/RS (accessed on 31 May 2022). On this open source website, we have introduced and explained the code in detail in separate documents for the analysis of vegetation index and the research of different classifications.

**Conflicts of Interest:** The authors declare no conflicts of interest.

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
