# Peer review of "The Classification Method Study of Crops Remote Sensing with Deep Learning, Machine Learning, and Google Earth Engine"

_remotesensing, doi:10.3390/rs14122758_

Round 1

Reviewer 1 Report

Section 3.3.4 must be explained in detail. Feature extraction, Random Forest, Feature Filtering modules should be explained. Their parameters must be introduced with their mathematical formulas or within a pseudocode.

Training of the deep learning layers should be explained in more detailed. The training method should be compared with approaches such as; training from scratch and transfer learning. Different deep learning structures and optimization functions should be compared. 

The Google Earth Engine and the Colab code of the developed framework can be shared with the audience. 

Reviewer 2 Report

This is an improved version of an already good paper.

Reviewer 3 Report

Dear authors, I just quickly went through the manuscript several days ago, and I found the author addressed almost all the comments, everything looks fine to me.

Reviewer 4 Report

Thanks the authors have addressed all issues. However, I still have some minor comments:

--> I recommend Fig. 7(b) can be ranked by feature importance instead of the feature name.

--> I still think the study area is too small to evaluate the effectiveness and robustness of your method. Could you please consider to extend your study area? For example, select another region, and discussion the transferability or the generalization of your model.

Some referecences maybe helpful for you:

Aptoula, E. (2021, June). Weed and Crop Classification with Domain Adaptation for Precision Agriculture. In 2021 29th Signal Processing and Communications Applications Conference (SIU) (pp. 1-4). IEEE.

Zheng, J., Fu, H., Li, W., Wu, W., Zhao, Y., Dong, R., & Yu, L. (2020). Cross-regional oil palm tree counting and detection via a multi-level attention domain adaptation network. ISPRS Journal of Photogrammetry and Remote Sensing167, 154-177.

Hamrouni, Y., Paillassa, E., Chéret, V., Monteil, C., & Sheeren, D. (2021). From local to global: A transfer learning-based approach for mapping poplar plantations at national scale using Sentinel-2. ISPRS Journal of Photogrammetry and Remote Sensing171, 76-100.

Round 2

Reviewer 1 Report

Thanks to authors for making required updates and providing a clear list of changes in the new manuscript.

This manuscript is a resubmission of an earlier submission. The following is a list of the peer review reports and author responses from that submission.

Round 1

Reviewer 1 Report

The submitted topic is well investigated in the literature in past 3 decades. Many statistical, probabilistic, computer vision, machine learning and AI based approaches are proposed and most of them are fully automated including their parameter selection steps. The submitted manuscript therefore does not provide extra mathematical advancements nor it solves a new challenge. The procedure relies on high amount of parameter selection with human intuition. The repeatability chances of the given science for different regions and data are very low. Thus, I do believe that the manuscript is not suitable for a journal. I recommend authors to show their experiments and results in a suitable conference instead.

Reviewer 2 Report

The demonstrated superiority of RF + DNN is convincing, but not surprising.  Two good overlapping but not identical classifiers combined should be better than any single ones. Nevertheless, the use of the cloud platforms in addition to the classifier evaluations makes this an especially interesting paper.

One minor question: why are there quotation marks on lines 232 and 234?

Reviewer 3 Report

Review of “The classification method study of crops remote sensing with deep learning, machine learning, and google earth engine

The manuscript entitled “The classification method study of crops remote sensing with deep learning, machine learning, and google earth engine” demonstrated a set of convenient tools for the research community to perform analysis regarding big/complex data sets. Also, the research reported in the manuscript compared the performance of some basic but popular and widely used machine learning methods in the aspect of the classification.  

Overall, this paper is well written. However, there is still some information missing that makes the manuscript vague and contently weakens the significance of the paper. A major/semi-major revision is recommended before publication. The following are some of the points when I went through the paper in detail.

It will be better for the author to give a mathematical introduction or have a specific section for the statistics used in this paper (GINI, OA, KC, etc.). For example, the author needs to explain what is GINI coefficient, why a parameter that is intended to represent the income inequality or the wealth inequality is selected to evaluate part of the performance for image classification?

The description of the data used in the training and validation process is vague. The author should provide a straightforward number for the amount of data used in the research.

  • How many days of data is obtained?
  • If pixel by pixel training was conducted, how many points does the training data set have?
  • What is the size of the ground truth data set?
  • What is the input data for each model used? What is the output? It will be good to have table or figure to summarize it instead of letting the reading themselves to search it between paragraphs. It will also be good to put these information onto each of the diagram of the model structure

Line 109 2.2 Data source and processing. The author highlighted the convenience of the Google platform for data processing. However, more detailed information should be provided by the author regarding the data they used in this research and the method Google platform applied to make the manuscript more reader-friendly.

  • What is the temporal and spatial coverage of the Sentinel -1 and -2?
  • What is the collocated method used to line up Sentinel -1 and -2, and what is the uncertainty of the collocated method that the Google platform used for the data processing of Sentinel -1 and -2?
  • What grid does the data set use? Sinusoid grid? UTM grid? Climate Modeling Grid?

Line 138, Filed survey data should be Field survey data?

Line 143, Does the Filed survey act as the ground truth in all the classification methods? Are the 300 samples the total amount of the ground truth data set? Is this big enough to drive a statistical stable deep neural network? This piece of information is crucial since DNN and other higher-order ML tools do have some sample amount dependence

Line 171, it would be more convenient for the author to provide the wavelength information of each band used, band number can seldomly help the reader to comprehend what information carried in that band.

Line 196 3.2. Crops phonological analysis. Is this part related to figure 6? If so, it will be good to point out, or simply consider the modify the structure a little bit to merge this part with the 4.1.

Line 294, it is obvious that different land surface types can have different spectral reflectivity signatures? The problem is how significant those signatures can be extracted to achieve the classification. It will be more straightforward to provide the correlation heatmap and the feature scores output from your RF model, this might help to highlight your point.

Reviewer 4 Report

This paper is a comparative study, which employs RF, OO, DNN, and RF+DNN with GEE and Google Colab to conduct crop extraction and classification. Sentinel-1 and Sentinel-2 data are used to generate vegetation index, band information and polarization. The results show that RF+DNN achieves the best accuracy. Here are some comments:

  1. Please state more about your work’s main motivation and contribution clearly. If you want to discuss machine learning/deep learning brings efficiency and high accuracy in crop extraction and classification, you need to state more about previous traditional methods for crop extraction and classification, i.e., manual extraction. But in fact there already exist many deep learning methods for crop extraction and classification. If you want to highlight the advantages of cloud platforms, you should also do more comparative work with the scenario where you work from the local. If your main contribution is “use Sentinel-1 and Sentinel-2 multi-source data on the cloud platform to construct vegetation index, band information and polarization”, then please focus on it and pay less attention to other aspects. To me, your motivation and contributions are somehow ambiguous.
  2. You could conduct your RF and OO experiments on the Colab platform as well. Different platforms may affect the performance of models, though deep learning methods definitely outperform the traditional machine learning methods. It’s better to compare all methods on the same platform.
  3. The related work’s review is not enough. Please show more traditional methods and deep learning methods in crop extraction and classification.
  4. Please pay attention to some spelling errors and grammatical mistakes, such as “2.3 Filed survey data”.
  5. I think you should present more about the methods. More pictures should be added to show the feature space construction. How do the vegetation index feature and polarization feature look like? Also In Fig. 12, it’s better if you put the ground truth label together with different results.
  6. Some references are recommended to add in this paper:

Hu, X., Wang, X., Zhong, Y., & Zhang, L. (2022). S3ANet: Spectral-spatial-scale attention network for end-to-end precise crop classification based on UAV-borne H2 imagery. ISPRS Journal of Photogrammetry and Remote Sensing183, 147-163.

Zheng, J., Fu, H., Li, W., Wu, W., Yu, L., Yuan, S., ... & Kanniah, K. D. (2021). Growing status observation for oil palm trees using Unmanned Aerial Vehicle (UAV) images. ISPRS Journal of Photogrammetry and Remote Sensing173, 95-121.

Pott, L. P., Amado, T. J. C., Schwalbert, R. A., Corassa, G. M., & Ciampitti, I. A. (2021). Satellite-based data fusion crop type classification and mapping in Rio Grande do Sul, Brazil. ISPRS Journal of Photogrammetry and Remote Sensing176, 196-210.